# Polaron spectroscopy of interacting Fermi systems: Insights from exact diagonalization

Ivan Amelio⋆ and Nathan Goldman

Center for Nonlinear Phenomena and Complex Systems, Université Libre de Bruxelles,
CP 231, Campus Plaine, B-1050 Brussels, Belgium

⋆ .amelio@ulb.be

## Abstract

Immersing a mobile impurity into a many-body quantum system represents a theoretically intriguing and experimentally effective way of probing its properties. In this work, we study the polaron spectral function in various environments, within the framework of Fermi-Hubbard models. Inspired by possible realizations in cold atoms and semiconductor heterostructures, we consider different configurations for the background Fermi gas, including charge density waves, multiple Fermi seas and pair superfluids. While our calculations are performed using an exact-diagonalization approach, hence limiting our analysis to systems of few interacting Fermi particles, we identify robust spectral features supported by theoretical results. Our work provides a benchmark for computations based on mean-field approaches and reveal surprising features of polaron spectra, inspiring new theoretical investigations.



## 1   Introduction

A mobile impurity immersed in a many-body background represents a paradigmatic setting of many-body physics. Historically, Landau and Pekar were the first to discuss the renormalized mobility of an electron interacting with the phonons of a polar crystal, and they coined the emerging quasi-particle *polaron* [1].

Following the achievements in the precise preparation and control of ultracold atomic mixtures, there has been a renewed interest in studying polarons at a fundamental level [2–5]. Indeed, immersing a single mobile impurity in a non-interacting or weakly-interacting background already represents a fully fledged many-body problem. So far, ultracold atom experiments have been able to probe the attractive and repulsive branches of standard, weakly interacting Fermi [6] or Bose [7,8] polarons, using radio-frequency spectroscopy. Such measurements are complicated by the metastability of atomic gases under three-body recombination and by the challenge of independently tuning the interactions within the background and those involving the impurity, so that polaron studies have yet to be extended to the case of strongly correlated backgrounds. It is nonetheless worth mentioning recent measurements of the so-called "magnetic polaron" in doped Hubbard anti-ferromagnets [9], of Nagaoka polarons arising from kinetic magnetism in triangular lattices [10, 11], and of the dressed, quantum-statistics-dependent interactions between polarons [12].

In parallel, the last years have witnessed the thriving of 2D materials, such as graphene and transition metal dichalchogenide (TMD) semiconductors [13]. Contrary to cold atoms, in TMD heterostructures experimentalists have access to few observables only, and optical polaron spectroscopy [14], where resonantly injected excitons are used to probe the state of the TMD material, is now daily used. More precisely, this method relies on the exciton being dressed by the electronic excitations, with the exciton-electron scattering properties being determined by the binding energy of the trion, which is the bound state of an exciton with an electron. Remarkably, TMD bilayers hosting moiré potentials have emerged as an ideal platform to simulate Fermi-Hubbard and extended Fermi-Hubbard physics [15,16]. Transport or optical signatures of several exotic phases of matter have already been reported, including Wigner crystals [17,18], charge density waves [19,20], excitonic insulators [21–23], superconductivity [24,25] and anomalous quantum Hall states [26]. However, the very large trion binding energy with respect to the moiré-electronic scales may have limited the number of features observable in polaron spectra, as we will further illustrate in this work.

Given the hardness of the many-body problem, polarons in strongly interacting many-body backgrounds have been theoretically considered in a few special settings only or under very rough approximations. We mention attempts in the context of fractional Chern insulators [27, 28], Fermi superfluids along the BEC-BCS crossover [29,30], excitonic insulators [31], Mott and charge transfer insulators [32], kinetic magnetism [33,34], Holstein polarons in Luttinger liquids [35], together with a few related works on the dressing of optical excitations in Mott insulators [36,37] and fractional quantum Hall systems [38]. Diffusion Quantum Monte Carlo can be used to study Bose [39] and Fermi polarons [40] in the intermediate correlation regime, but does not allow for the determination of the full spectral information.

In this work, we use exact diagonalization (ED) to tackle this challenging problem, exploring various configurations of the interacting Fermi gas. Specifically, we compute zero-momentum polaron spectra in lattice systems of a few interacting fermions, both in the spinless and in the spinful case, with different classes of repulsive or attractive interactions. The main drawback of ED is that this method is limited to very small system sizes. In spite of these finite-size effects, our results are qualitatively consistent when comparing between different lattices or, when possible, to other methods (e.g. using the Chevy Ansatz within a mean-field theory for the many-fermion background). In this sense, we believe that our ED approach provides solid results, which allow to benchmark approximate methods but also provide novel insights. Moreover, studying how the growth of many-body correlations scale with the system size is an interesting topic on its own [41, 42].

This manuscript is structured as follows: in Section 2, we illustrate the ED method and benchmark it for an impurity immersed in a non-interacting Fermi sea. In Section 3, we discuss polaron spectra in the presence of strong charge density wave correlations for a system of spinless fermions with long range repulsion. In Section 4, we tackle the spinful case with a spin-dependent impurity-fermion interaction, and we study the fate of two-body and three-body absorption lines in the presence of contact repulsive interactions between the fermions. In Section 5, we consider polaron spectroscopy of fermionic superfluids, in the attractive Fermi-Hubbard model, both for balanced and nearly-balanced populations of the two species. Finally, we draw our conclusions and outline future directions in Section 6.

## 2 Polaron spectra from exact diagonalization

### 2.1 General framework

This work explores the problem of an impurity immersed in a system of spin 1/2 fermions, described by a Hamiltonian of the form

$$H = H_{ff} + H_I + H_{fI}. \tag{1}$$

Here $H_{ff}$ denotes the interacting fermionic background, described by a generalized Fermi-Hubbard Hamiltonian

$$H_{ff} = -t_f \sum_{\langle i,j \rangle, \sigma} c_{i\sigma}^\dagger c_{j\sigma} + \frac{1}{2} \sum_{i,j,\sigma,\sigma'} V_{ij} c_{i\sigma}^\dagger c_{j\sigma'}^\dagger c_{j\sigma'} c_{i\sigma}, \tag{2}$$

where $c_{j\sigma}^\dagger$ denotes the creation operator of a fermion of spin $\sigma \in \{\uparrow, \downarrow\}$ at lattice site $j = (x, y)$, $t_f$ is the hopping constant and $V_{ij}$ denotes the matrix elements of a general (possibly long range) interaction between the fermions.

The kinetic energy for free impurities is $H_I = -t_{\text{im}} \sum_{\langle i,j \rangle} a_i^\dagger a_j$, where $a_j^\dagger$ denotes the creation operator of the impurity and $t_{\text{im}}$ its hopping rate. In this work, we will limit ourselves to contact impurity-fermion interactions

$$H_{fI} = \sum_{j,\sigma} U_\sigma c_{j\sigma}^\dagger c_{j\sigma} a_j^\dagger a_j. \tag{3}$$

Notice that the coupling constant $U_\sigma$ depends on the spin of the fermion. This is a natural choice both in atomic systems, where Feshbach resonances are spin-dependent [43], and in solid-state spectroscopy, where the trion binding energy depends on the polarization of the probe excitons [44–46] (e.g. in molybdenum based TMD monolayers, spin-triplet trions are typically unbound).

Describing a Fermi polaron in a non-interacting Fermi sea is already a fully fledged many-body problem, which cannot be solved exactly [2–4]. Here we commit ourselves to the study of models of interacting fermions, such that accessing the ground state $|f_0\rangle$ of $H_{ff}$ already represents a formidable task. We address this problem using ED, which consists in expressing the Hamiltonian as a sparse matrix in the real space basis [47–50], and in using the Lanczos method [51] to determine the ground state.

ED gives also access to the ground state $|\Psi\rangle$ of the full impurity-fermion Hamiltonian $H$. However, the most natural observable in both ultracold atom [52] and solid-state experiments [14] is provided by the spectral function of the impurity. Our main goal is thus to compute polaron spectra, which experimentally corresponds to the following protocol: first, the many-fermion system is prepared in its ground state in the absence of the impurity, or the fermion-impurity interaction is switched off by preparing the impurity in some hyperfine level; then, the impurity is resonantly injected, or its internal state is resonantly flipped to a hyperfine level characterized by a sizable fermion-impurity interaction; the final observable is the frequency-resolved absorption curve of the resonant excitation. Mathematically, the polaron spectral function is defined as

$$A(\omega) = -2\mathrm{Im}\langle \Psi_0 | \frac{1}{\omega - H + E_0} | \Psi_0 \rangle \,, \qquad (4)$$

where $|\Psi_0\rangle$ is the ground state of the fermion-impurity decoupled Hamiltonian $H_{ff} + H_I$, with energy $E_0$, while $H$ is the full Hamiltonian with finite fermion-impurity coupling. Since generally $|\Psi_0\rangle = a^\dagger_{k=0}|f_0\rangle$, one can make a link with optical spectroscopy in TMDs, where an exciton is injected with very small momentum. The spectral function is also related via Fourier transform to the overlap $S(t) = \langle \Psi_0 | e^{-i(H - E_0)t} | \Psi_0 \rangle$ following a quench of the impurity-fermion interaction. Notice that the spectral function (4) satisfies the normalization $\int_{-\infty}^{\infty} \frac{d\omega}{2\pi} A(\omega) = 1$. In plotting the spectra, the lines are artificially broadened by replacing $\omega \to \omega + i\gamma$.

A crucial technical remark concerns the fact that, even though the full spectrum of $H$ cannot be obtained for large sparse matrices, the spectral function can still be reliably and efficiently obtained [48, 51, 53]. The trick consists in constructing the Krylov space of dimension $M_{\mathcal{K}}$ by applying $M_{\mathcal{K}}$ times the full Hamiltonian $H$ to the decoupled ground state $|\Psi_0\rangle$. The Hamiltonian in this space is represented by a tridiagonal matrix, for which the resolvent of the first entry can be conveniently computed by recursion and expressed as a continued fraction. It can be proven that this approach captures exactly the first $2M_{\mathcal{K}} + 1$ moments of the spectral function.

In this work, we restrict ourselves to nearest-neighbour hopping without complex phases and we consider two-dimensional square or triangular lattices. We will be limited to very small system sizes such that finite-size effects represent a major concern; however, in the following, we will show and argue that one can still extract solid and useful qualitative insights using this approach. In our philosophy, this method naturally needs to be complemented with other approaches, such as Chevy Ansatz computations [54, 55]. For instance, Section 5 focuses on a polaron in a Fermi superfluid, which is a good example of how ED can confirm a highly nontrivial feature previously reported using mean-field calculations complemented by the Chevy Ansatz [30, 31], an approach reviewed in Appendix D.

An insidious consequence of finite-size effects is that the ground states $|\Psi_0\rangle$ and/or $|\Psi\rangle$ may not be invariant under the action of the spatial symmetries of the model. For instance, the ground state of the non-interacting system with 2 identical fermions (i.e. $N_\uparrow = 2$, $N_\downarrow = 0$, $V_{ij} = 0$) consists of a superposition of states in the form $c^\dagger_{k=0\uparrow} c^\dagger_{k\neq0\uparrow}|0\rangle$, which is not a zero momentum eigenstate of the total momentum. In the following, we restrict ourselves to the sector of zero total momentum and require that $|\Psi_0\rangle = a^\dagger_{k=0}|f_0\rangle$, i.e. that the fermion and impurity momentum before quenching the impurity-fermion interaction are separately zero:

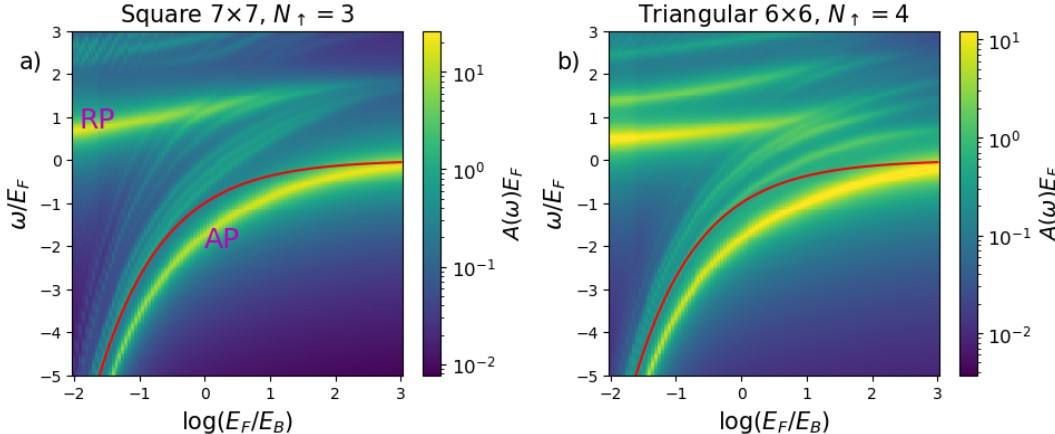

Figure 1: Polaron spectra on top of a spin-polarized non-interacting Fermi sea. Two ED calculations are compared, on a $7 \times 7$ square lattice with $N_\uparrow = 3$ fermions (left panel) and on a $6 \times 6$ triangular lattice with $N_\uparrow = 4$ (right panel). The red line represents $-E_B$, while AP and RP indicate the attractive and repulsive polaron branches, respectively. The spectral function intensity is color-coded in a logarithmic scale.

this imposes a strong constraint on the number of particles that one can fit into the system. In the case of crystalline phases, one should also respect the commensurability of the crystal with respect to the size of the system.

## 2.2 Benchmarking Fermi polarons

In this subsection, we consider the simplest scenario of an impurity immersed in a non-interacting Fermi sea of spin-polarized fermions ($N_\downarrow = 0$ for definiteness).

The fermion-impurity binding energy in the vacuum $E_B > 0$ is related to the coupling $U_\uparrow$ appearing in $H_{fI}$ by the Lippmann-Schwinger equation [56]

$$\frac{1}{U_\uparrow} = \frac{1}{L_x L_y} \sum_k \frac{1}{-E_B - \epsilon_k^f - \epsilon_k^I} \, , \tag{5}$$

with $L_x, L_y$ the number of lattice sites in the two independent spatial directions and $\epsilon_k^f, \epsilon_k^I$ denoting the free fermion and impurity dispersions, respectively. We define the Fermi energy through the formula $E_F = \frac{2\pi\hbar^2}{m_\uparrow} n$, which holds in continuous infinite systems, where $n = N_\uparrow / A$ is the density. Here we defined the area of the system $A$, where, assuming unit distance between adjacent lattice sites, $A = L_x L_y$ for a square lattice and $A = \frac{\sqrt{3}}{2} L_x L_y$ for triangular lattices; furthermore, for the effective fermion mass $m_f$, one can use the correspondence $\frac{\hbar^2}{2m_f} = t_f$ on a square lattice and $\frac{\hbar^2}{2m_f} = \frac{3}{2} t_f$ on a triangular one, as it can be deduced from the curvature of the single particle dispersion at small momentum.

Rescaling the energies by $E_F$, and as far as properties such as the positions of the repulsive and attractive polaron peaks or the oscillator strength transfer are concerned, we find that calculations performed using different sizes and lattices yield consistent results, also compatible with Chevy Ansatz computations [54,55,57,58]. We note that other properties, such as details of the molecule-hole continuum, the broadening of the repulsive polaron or some very weak high-energy peaks, were found to strongly depend on the size of the system, clearly showing signatures of the modes discretization. Two examples are shown in Fig. 1, where we compare Fermi polaron spectral function for a $7 \times 7$ square lattice with $N_\uparrow = 3$ fermions (left panel) to

the case of a $6 \times 6$ triangular lattice with $N_\uparrow = 4$ (right panel). The red lines mark the vacuum impurity-fermion binding energy $-E_B$, while the magenta AP and RP labels indicate the attractive and repulsive polaron branches, respectively. In these plots we used a small linewidth $\gamma = 0.1 E_F$ and a logarithmic color scale to highlight the fine structure of the molecule-hole continuum, which depends on the details of the computation.

We devote Appendix A to a more detailed comparison between different lattice sizes and number of particles, as well as to showing convergence in the Krylov space dimension $M_{\mathcal{K}}$.

## 3 Charge density waves

In this section, we will be interested in the physics of fermions with repulsive finite-range interactions. For definiteness, we will consider spin-polarized systems with Coulomb repulsion $V_{ij} = \frac{V_0}{|\mathbf{r}_i - \mathbf{r}_j|}$, where $V_0$ is the coupling constant and $\mathbf{r}_i$ denotes the position of the $i$-th lattice site. Given the underlying lattice structure and the finite filling factors used in the following, we expect our results to qualitatively hold for generic finite-range repulsive potentials.

Polaron spectroscopy has already been used in experiments to detect Wigner crystals in TMD monolayers [17] and bilayers [18] without a sizable moiré potential. The main signature of the presence of the Wigner crystal is a weak umklapp line, which departs from the repulsive polaron peak with a splitting that scales with doping[1] like $\propto \frac{\hbar^2}{2 m_X a_W^2}$, with $m_X$ the mass of the exciton and $a_M$ the lattice constant of the Wigner crystal. These experiments can be modeled assuming that the Wigner crystal is a static external potential that scatters the probe exciton.

While Wigner crystals are formed in continuum systems and spontaneously break a continuous translational symmetry, our ED approach is limited to finite-size lattice models, and we will refer to states with strong crystalline fluctuations as "charge density waves". Because of the finite size, the discrete translational symmetry cannot be spontaneously broken and one needs to inspect the density-density correlator $\langle f_0 | n_i n_j | f_0 \rangle$, with $n_i = c_{i\uparrow}^\dagger c_{i\uparrow}$, in order to monitor the crossover from a Fermi liquid to a charge density wave.

Impurities in lattice systems in the presence of strong repulsion have been considered as well in the TMD literature. For instance, the umklapp peak was used to reveal the onset of charge incompressibility of correlated electrons in moiré bilayers [59], even without the breaking of the discrete translational symmetry. The optical signatures of generalized trions in a moiré system displaying charge density waves at fractional filling were very recently reported in [60].

We computed the polaron spectrum for a $6 \times 6$ square lattice with $N_\uparrow = 4$ fermions, a number commensurate with the formation of a crystal of lattice constant $a_{CDW} = 3 a_0$. We fix the strength of fermion-impurity interactions according to $E_B/E_F \simeq 2.2$ and scan a broad range of values of $V_0$; as it is usually done for Wigner crystals, we express the strength of the interactions as $\kappa = \frac{V_0}{a_{CDW} E_F}$, i.e. the ratio of the typical interaction to kinetic energy. The results are shown in Fig. 2. In panel (a) the polaron spectrum is depicted as a function of $\kappa$. The color mesh is in linear scale, while in panel (b) we plot exactly the same data in log scale. In panels (c) and (d), computed along the vertical dotted slices of panel (a), we display the density-density correlator $\langle f_0 | n_{(0,0)} n_{(x,y)} | f_0 \rangle$ at small and large $\kappa$, respectively. These two plots illustrate the building up of crystalline correlations along the crossover from Fermi liquid to charge density wave.

Let us now focus on the features visible in panels (a) and (b). To start with, the attractive polaron (AP) experiences a redshift for increasing $\kappa$. We attribute this feature to the renor-

---

[1]In borrowing from semiconductor physics the term doping, we mean the density of active electrons in conduction bands, which in TMDs can be controlled via electrical gates. In our lattice model, it just corresponds to the density of fermions.

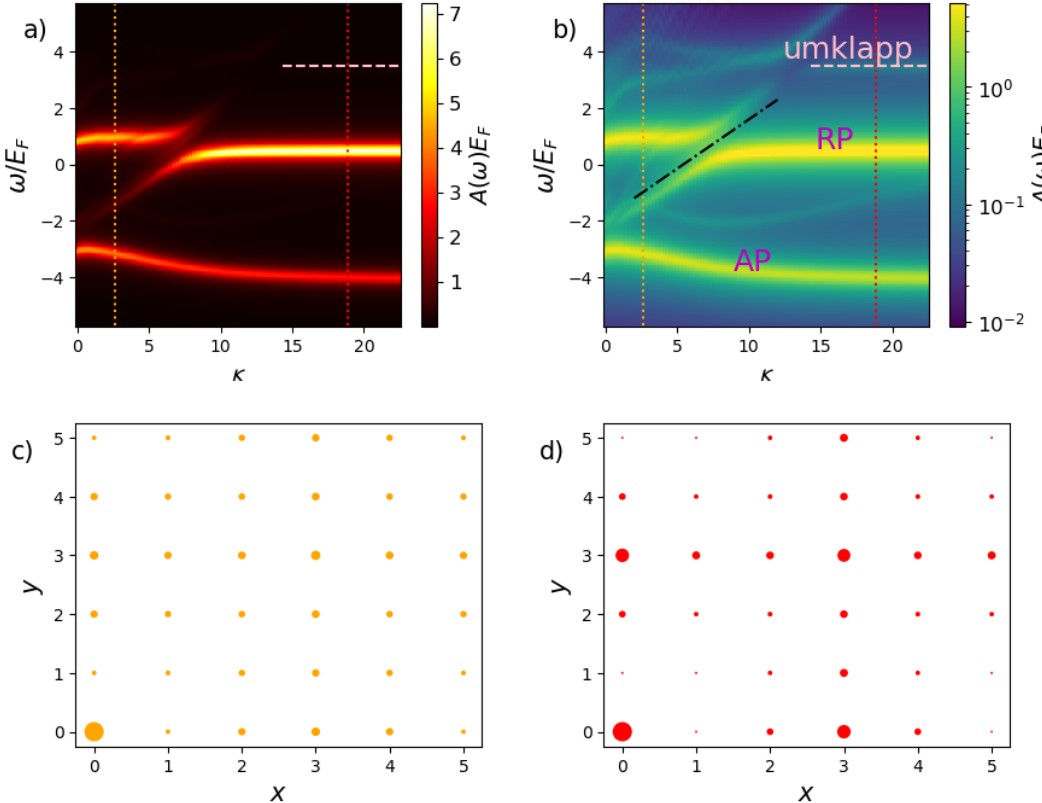

Figure 2: Polaron spectroscopy of the crossover from non-interacting Fermi sea to charge density wave, for fixed $E_B/E_F \simeq 2.2$. (a) Polaron spectrum for increasing values of the interactions. Horizontal pink dashes indicate the position of the very weak umklapp peak (visible only from panel (b)). The vertical orange and red dots correspond to panels (c) and (d), respectively. (b) Same as (a) but in logarithmic color scale, in order to highlight the weaker features. The black dash-dots are a guide to the eye for the line that blueshifts with $V_0$ and generates the avoided crossing with the repulsive polaron branch RP. AP indicates the attractive polaron. Panels (c) and (d) report the density-density correlator $\langle f_0|n_{(0,0)}n_{(x,y)}|f_0\rangle$ at small and large $V_0$, respectively. The lattice is a $6 \times 6$ square with periodic boundary conditions and $N_\uparrow = 4$ fermions.

malization of the fermion mass, resulting in a larger binding energy; in particular, in the limit $V_0 \to \infty$ all the fermions are perfectly correlated, so the impurity binds to an object of total mass $N_\uparrow m_f$. Then, we have highlighted by the pink dashed line the umklapp peak [17], well defined at large $V_0$. This can be thought of as a scattering state of the impurity with the fermionic crystal and occurs at an energy essentially determined by the folding of the free impurity dispersion in the first Brillouin zone of the crystal, i.e. in a square lattice at wavevector $2\pi/a_{CDW}$. In Appendix B we have indeed verified that this peak behaves like expected.

Finally, the most intriguing and original feature is the avoided crossing occurring for comparable values of the binding energy $E_B$ and the effective interaction strength $V_0/a_{CDW}$. Here we speculate on the origin of this effect, which could be further studied and confirmed in the future using the Chevy Ansatz. In a Hartree-Fock picture, the formation of a charge density wave is described in terms of Bloch waves on top of the Hartree-Fock potential, with the same periodicity as the charge density wave and determined self-consistently by filling the lowest Bloch band. When $V_0$ is increased, the gap between the lowest and second band scales ap-

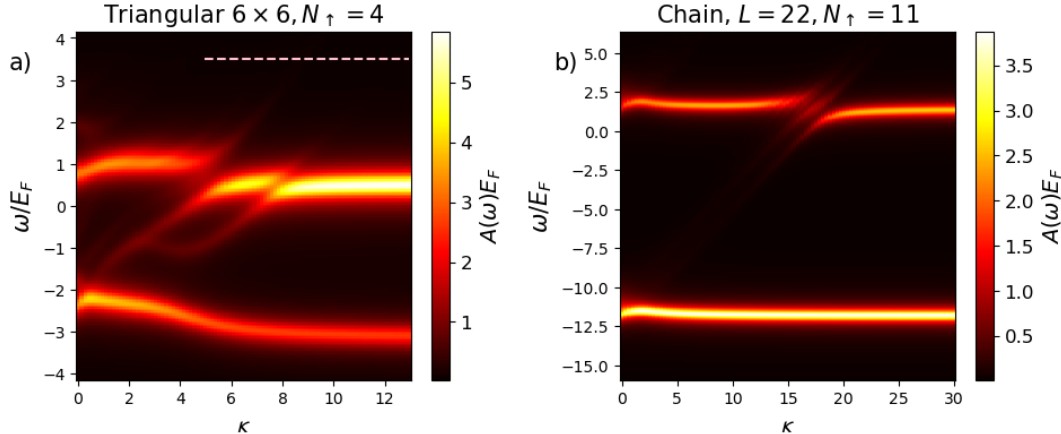

Figure 3: Polaron spectrum for increasing values of Coulomb repulsion on a triangular lattice (left) and on a chain (right), respectively. Pink dashes indicate the position of the umklapp peak (not visible on this scale), which is absent in the 1D system.

proximatively linearly; moreover, the low-lying bands also become flatter and flatter. It seems then likely that the repulsive polaron (RP) hybridizes with the state made of the bound impurity plus an excitation of the neutral mode corresponding to the interband excitation of one Hartree-Fock quasi-particle; the dashed-dotted line in panel (b) is a tentative guide to the eye for the energy of this candidate state, which blueshifts linearly with $V_0$. Notice that a similar feature was observed for a homobilayer model with local repulsion, in a ladder computation using the electron Green's function computed in DMFT, and was attributed to doublons [32]. An even more characteristic feature is observed on a triangular lattice, as we show in Fig. 3(a). Understanding this peculiar behavior is also the subject of ongoing efforts.

A piece of evidence that these results are not just an artifact of finite-size effects comes from the study of one-dimensional chains, for which it is possible to reach a reasonably large number of sites $L = 22$ with $N_\uparrow = 11$ fermions. In 1D, we define the Fermi energy as $E_F = 2\pi t_f n^2$. As one can see in Fig. 3(b), the avoided crossing is also found in such a long chain. Interestingly, no umklapp peak is found in this 1D case.

To conclude this section, we comment on the relation of our results with existing experiments in TMDs. The avoided crossing, in particular, has not been observed in the charge density waves and Wigner crystals reported so far in the literature. This is due to the fact that the trion Bohr radius is much smaller than a moiré cell, or in other words, the trion binding energy is large compared to the electronic many-body gap. The avoided crossing occurs when the two energies are comparable, and this could be made possible by engineering trions with a smaller binding energy in heterostructures with strong moiré potential. Such trions may be accessible in multilayer systems, with the excitons and the fermions mainly localized in two different layers.

# 4 Impurity interacting with two Fermi seas

While polaron spectra of molybdenum-based TMDs are essentially consistent with the Fermi polaron picture [14, 61], experiments in WSe₂ monolayers display a much richer structure [44, 62]. At small electron doping, two attractive polaron lines are visible together with the repulsive polaron; at high densities, instead, all the oscillator strength is taken by a low energy line, which redshifts with increasing density.

This difference originates from the fact that two identical electrons and a hole will generally not bind into a trion, as a consequence of the anti-symmetrization of the wavefunction [44–46]. In $MoS_2$ and $MoSe_2$ monolayers exciton formation involves the lowest conduction bands of quantum numbers $(K, \uparrow)$ and $(-K, \downarrow)$, so that right circularly polarized radiation can excite only the $(K, \uparrow)$ transition and the only visible trion will be the one formed from this exciton and an electron in $(-K, \downarrow)$. Here, $\pm K$ denote the corners of the first Brillouin zone and the arrows the physical out-of-plane spins. In $WSe_2$ monolayers, instead, the bright conduction bands $(K, \uparrow)$ and $(-K, \downarrow)$ lie higher in energy than the dark $(K, \downarrow)$ and $(-K, \uparrow)$ lower conduction bands. This means that the dark conduction bands get doped, while the electron forming an exciton comes from the bright conduction bands and can form a trion with either the $(K, \downarrow)$ and $(-K, \uparrow)$ electron, which is distinguishable by valley or spin.

This argument explains qualitatively the presence of two attractive polaron resonances. Some theoretical understanding of the crossover to a single line red-shifting with doping has been provided in [62, 63] based on variational wavefunction calculations. The first step is to realize that the attractive polaron is not really associated with a trion (exciton bound to an electron), but rather with a tetron, i.e. an exciton bound to an electron-pair formed out of the conduction band Fermi sea. At small doping, the phase space of the hole is limited to very small momenta and the tetron is basically a trion very loosely bound to a hole. In practice, to locate the two attractive polaron lines at small doping, one can compute the binding energy of the singlet and triplet trion. However, when the exciton can bind to two Fermi seas and at large doping, a so-called "hexciton" can be formed from the exciton and a particle-hole pair out of each of the two Fermi seas. The high doping condition ensures that the particle-hole pair is small and akin to a neutral object; otherwise, if the holes were highly delocalized, the exciton could not bind to two charged electrons.

This rich physics can, in principle, also be accessed in ultracold atom setups with three species [64, 65]; however, to our knowledge, no experiment has ever investigated the spectral properties of a polaron in two Fermi seas to this date.

Here, we apply our ED method to an impurity immersed in a background with two species of fermions. For the sake of dealing with only one parameter to tune the strength of repulsion, we consider the case of contact repulsive interactions with coupling $V_{ij} = V_{\uparrow\downarrow}\delta_{ij}$. We have checked that one obtains similar results in the presence of Coulomb terms, both inter- and intra-spin (contact interactions, instead, are only effective for fermions of different spin, due to the Pauli principle). To observe two attractive polaron peaks, we allow for spin imbalance of the contact fermion-impurity attraction $U_\uparrow \neq U_\downarrow$. In order to plot our results, we define the mean $U = \frac{U_\uparrow + U_\downarrow}{2}$ and the binding energy scale $E_B$ from the energy of the molecule[2] formed by the impurity and a fermion with coupling $U$, according to $\frac{1}{U} = \frac{1}{L_x L_y} \sum_k \frac{1}{-E_B - \epsilon_k^f - \epsilon_k^I}$. Since we cannot continuously change the density, we change the ratio $E_F/E_B$ by varying $U$.

Exact diagonalization results are plotted in Fig. 4 for a system of $N_\uparrow = N_\downarrow = 4$ fermions in a $4 \times 4$ triangular lattice (plots for the square lattice are very similar, see Appendix C). We choose a small spin asymmetry of $\frac{U_\uparrow - U_\downarrow}{U} = 0.3$ for the fermion-impurity interaction and we increase the fermion-fermion contact repulsion $V_{\uparrow\downarrow}$ through panels (a-c). Apart from the repulsive polaron peak, one can spot two attractive polaron resonances, denoted respectively $AP_\uparrow$ and $AP_\downarrow$ in panel (a), which are associated with spin up and spin down two-body bound states and which are predominant at small $E_F/E_B$; and a lower energy resonance $AP_{\uparrow\downarrow}$, corresponding to the three-body bound state of the impurity with both the spin up and spin down particles.

---

[2]Notice that such a fictitious molecule is just an abstract object, which we invoke to organize our results and does not correspond to any physical spectral line.

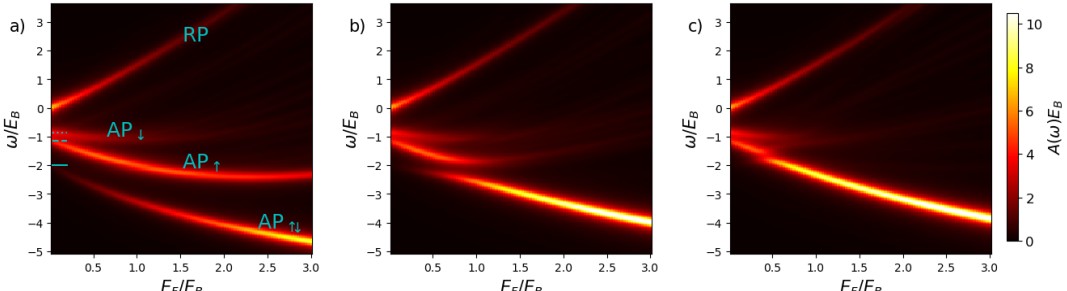

Figure 4: Polaron spectra for a spinful Fermi system with a slightly spin-asymmetric binding energy to the impurity. On the $x$-axis $E_B$ is decreased and panels (a-c) correspond to increasing contact repulsive interactions, $V_{\uparrow\downarrow} = 0, 2E_F, 4E_F$ respectively. The central panel is particularly reminescent of spectral measurements in WSe$_2$ monolayers. The main lines are labeled in panel (a): RP denotes the repulsive branch, AP$_\uparrow$ (AP$_\downarrow$) is the attractive polaron associated with the 2-body bound state between the impurity and the spin up (down) fermion, while AP$_{\uparrow\downarrow}$ comes from the 3-body bound state. The cyan dotted, dashed and solid lines at small $E_F/E_B$ indicate, respectively, the AP$_\downarrow$, AP$_\uparrow$ and three-body bound states in vacuum. Computation for a $4 \times 4$ triangular lattice with $N_\uparrow = N_\downarrow = 4$ fermions.

The identification of these lines is supported by the calculation of the two- and three-body binding energies in vacuum, which recover the many-body lines in the limit of small $E_F/E_B$ (cyan lines in panel (a)).

In the absence of the repulsive interactions between the fermions, $V_{\uparrow\downarrow} = 0$, all these three lines are simultaneously visible, with a slow oscillator strength transfer to the three body resonance for increasing $E_F/E_B$. We explain this effect by noting that, for the three-body line, the oscillator strength should scale with the square of the density, while for the usual two-body attractive polarons it scales linearly in the density. In other words, at large $E_B$ the three-body state has a very small radius and its wavefunction is very different from the ground state without impurity.

When the repulsive interaction $V_{\uparrow\downarrow}$ is turned on, there seems to be an avoided crossing between the lowest two-body line AP$_\uparrow$ and the three-body line AP$_{\uparrow\downarrow}$. This comes from the fact that, at first order in perturbation theory, the three-body state blueshifts linearly with $V_{\uparrow\downarrow}$, since the spin up and down fermions are bound together by the interaction mediated via the impurity. Interestingly, panel (b) reminds of the experimental data from WSe$_2$ monolayers [62]. The splitting of the avoided crossing decreases with $V_{\uparrow\downarrow}$, see panel (c). Notice that, since the vertical axis is in units of $E_B$, the redshift of the lines with $E_F/E_B$ is due to the well-known shift of the attractive polaron with density, which is linear for small $E_F/E_B$ [66].

Our method is somehow complementary to the variational computation of [62,63], which is effectively a few-body computation, where the many-body background is traced out and enters via an effective static screening potential. Also, the variational method is well suited to compute the ground state energy and oscillator strength, but it is difficult to get information about the full spectrum using that method. This makes our ED approach particularly interesting to study the crossover between the two-body and three-body lines as a function of repulsive interactions.

# 5 Fermi superfluids

In this section, we will be dealing with polaron spectroscopy of Fermi superfluids, with the pairing of spin up and down fermions. The fermionic Hamiltonian we consider is essentially an attractive Hubbard model, with contact inter-spin interactions $V_{ij} = \delta_{ij} V_{\uparrow\downarrow}$, where $V_{\uparrow\downarrow} < 0$ is related to the binding energy $E_{\text{pair}}$ of a fermionic pair in vacuum via the Lippmann-Schwinger equation

$$\frac{1}{V_{\uparrow\downarrow}} = \frac{1}{L_x L_y} \sum_k \frac{1}{-E_{\text{pair}} - 2\epsilon_k^f} \, . \tag{6}$$

As to the fermion-impurity interaction, we allow for spin-dependent interaction $U_\uparrow \neq U_\downarrow$. As in the previous section, we define $U = \frac{U_\uparrow + U_\downarrow}{2}$ and the binding energy $E_B$ from the energy of the molecule formed by the impurity and a fermion with coupling $U$. In other words, $E_B$ and $U$ are related by $\frac{1}{U} = \frac{1}{L_x L_y} \sum_k \frac{1}{-E_B - \epsilon_k^f - \epsilon_k^I}$.

The motivation to study this setting comes from future experiments in both ultracold atoms and solid-state devices. On the ultracold atom side, the tunability of interactions via Feshbach resonances has allowed to observe the BEC-BCS crossover [67]. More recently, there has been some effort in implementing three-species Fermi mixtures [64,65], also inspired by analogies with the $SU(3)$ group relevant in high-energy physics for the theory of the strong interaction. Therefore, polaron experiments in Fermi superfluids will hopefully be realized in the near future. On the theory side, a Chevy Ansatz study of polaron formation in 3D Fermi superfluids with spin-symmetric $U_\uparrow = U_\downarrow$ interactions has been performed in [29], while in the fully spin asymmetric case $U_\downarrow = 0$ full spectra were reported in [30], in both 2D and 3D.

On the solid-state side, instead, we were particularly inspired by the possibility of optically probing the presence of pairing in putative excitonic insulator states found in recent transport experiments in TMD heterostructures [21–23,68]. In this case, a 2D Chevy Ansatz study was performed in [31], where the two fermionic species correspond to two different layers and fully pseudo-spin asymmetric interactions were considered. Remarkably, unbalanced electron-hole mixtures were very recently demonstrated [68–70].

## 5.1 Spin-balanced fermion populations

We used our ED approach to compute polaron spectra in a Fermi superfluid background. The results are reported in Fig. 5 for a $4 \times 4$ triangular lattice with $N_\uparrow = N_\downarrow = 4$. As detailed in Appendix C, similar results were obtained for for $N_\uparrow = N_\downarrow = 3$ fermions on a $5 \times 5$ triangular or square lattice. In Fig. 5, $U_\sigma$ is different but fixed in each panel, while $E_{\text{pair}}$ is scanned. Panels (a-c) correspond respectively to $\frac{U_\uparrow - U_\downarrow}{U} = 2, 1, 0$ and $E_B/E_F \simeq 0.3, 0.5, 0.5$. In other words, the interaction is fully spin asymmetric in (a), while it is symmetric in (c), and panel (b) lies in between. The solid cyan line, labeled $E_3$, represents the energy of the three-body bound state in vacuum, while the dashed-dotted line $E_3^*$ stands for the first three-body excited state.

The lowest line can be identified as the attractive polaron corresponding to the 3-body bound state. On the BCS side it is mainly the interaction with the impurity to provide the attraction, while one rather has a Fermi pair bound to the impurity on the BEC side. This follows from the fact that in the small $E_{\text{pair}}$ limit one basically has a Fermi polaron, while for large $E_{\text{pair}}$ a Bose polaron description is adequate. Unexpectedly, in the spectra this occurs not just as a redshift of a single line, but for large $E_{\text{pair}}/E_F$ the attractive polaron line broadens and develops a secondary peak on the high energy side. Also, there is a certain sensitivity to the finite size of the system, since the secondary peak appears on the lower energy side in the instances of Appendix C. Interestingly, this feature is completely missing in the Chevy Ansatz calculations detailed in Appendix D and it will be interesting to further investigate this in the

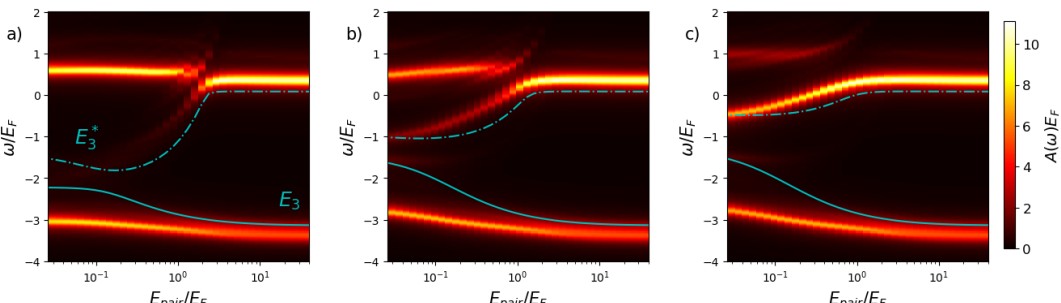

Figure 5: Polaron spectra for the Fermi superfluid, with $E_{\text{pair}}/E_F$ tuning along the BEC-BCS crossover. Panels (a-c) correspond to increasing spin-symmetry of the fermion-impurity interactions, namely $\frac{U_\uparrow - U_\downarrow}{U} = 2, 1, 0$. In the three panels we set $E_B/E_F \simeq 0.3, 0.5, 0.5$ respectively and $N_\uparrow = N_\downarrow = 4$ on a $4 \times 4$ triangular lattice. The cyan lines stand for the energies of the two lowest three-body bound states in vacuum, the ground state $E_3$ (solid line) and the first excited $E_3^*$ (dash-dotted). The pink arrows highlight the presence of a weaker peak below the main AP line at large $E_{\text{pair}}/E_F$.

future. We speculate that it may be due to the involvement of Goldstone-like excitations, which are not captured in the Chevy approach. Another explanation may be provided by some sort of molecule-polaron transition [71]. A more precise characterization is left as an open problem, but this phenomenon will constitute a good motivation and benchmark for improving on the current Chevy framework.

The other non-trivial feature visible in the spectrum of panel (a) is the avoided crossing on top of the repulsive branch for $E_{\text{pair}} \sim E_F$. This is perfectly consistent with the Chevy Ansatz prediction of [30,31], which we adapt in Appendix D to our lattice setting. The wavefunction of the state shifting with $E_{\text{pair}}$ was shown to have $2s$ symmetry, suggesting its relation to the Higgs mode of the superfluid on the BCS side and to the pair $2s$ excited state on the BEC side [30,31]. We remark that it makes a very strong argument that two completely different methods (i.e. Chevy Ansatz and ED) show the same feature, since the former is obtained by a mean-field BCS theory approximation followed by a variational restriction to the space of one quasi-particle pair excitations, while the latter is an exact method suffering from finite-size effects.

## 5.2 Spin-unbalanced fermion populations

Within ED one can also consider the case $N_\uparrow \neq N_\downarrow$, i.e. a homogeneous mixture with spin-unbalanced fermion numbers. This may be achieved in both ultracold mixtures and excitonic insulator setups, even though instabilities towards Fulde-Ferrell-Larkin-Ovchinnikov states or phase separation may strongly limit the available parameter region [72,73].

In Fig. 6, we present spectra for a $4 \times 4$ square lattice with $N_\uparrow = 4, N_\downarrow = 3$. We fix $E_B \simeq 0.5 E_F$ (where the Fermi energy is computed for spin up) and vary $E_{\text{pair}}$. Panels (a-c) correspond to different fermion-impurity interactions, namely $\frac{U_\uparrow - U_\downarrow}{U} = 2, 1, 0, -2$.

We hereby highlight some interesting features. First of all, the three-body attractive polaron line redshifts and slightly broadens for increasing $E_{\text{pair}}/E_F$, in analogy with the results shown in Fig. 5. Second, analogously to Fig. 5, an avoided crossing is well visible on the repulsive line in the panel (a) or (d) corresponding to $U_\downarrow = 0$ or $U_\uparrow = 0$, respectively. Finally, in the BEC limit one has three tightly bound fermionic pairs plus one spin up unpaired fermion (for this choice of particle numbers $N_\uparrow = 4, N_\downarrow = 3$). Then, for $U_\uparrow < 0$ the impurity can bind

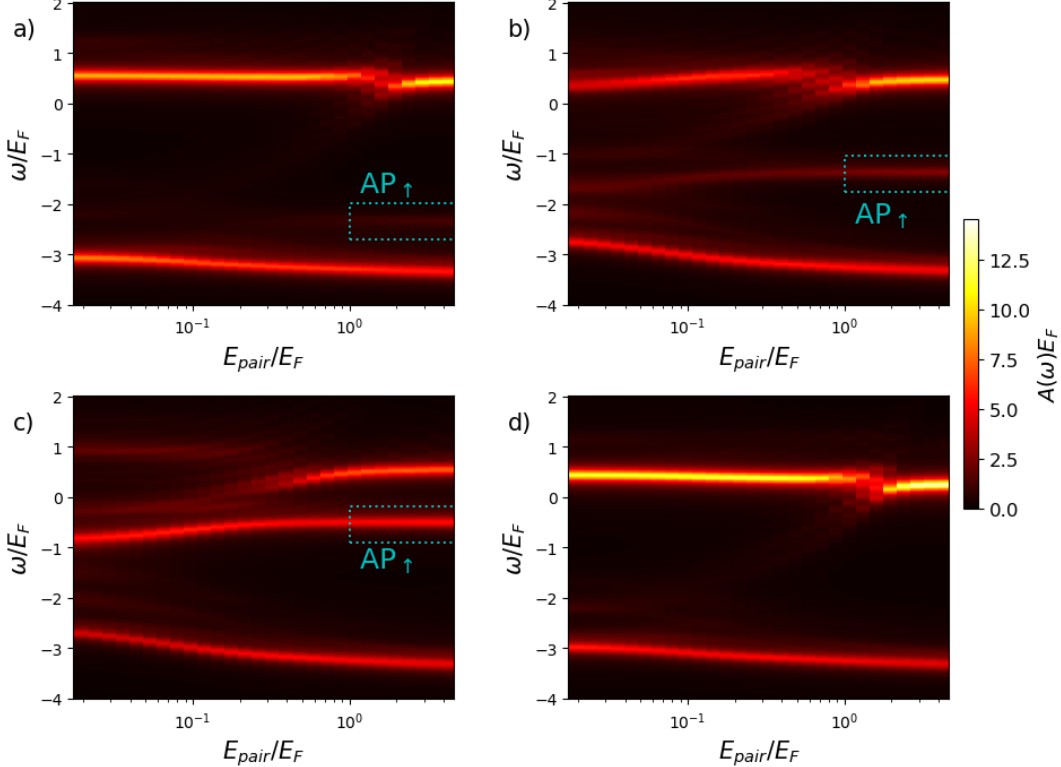

Figure 6: Polaron spectra for spin-density unbalanced Fermi superfluid, with $E_{\text{pair}}/E_F$ tuning along the BEC-BCS crossover. Panels (a-d) correspond to $\frac{U_\uparrow - U_\downarrow}{U} = 2, 1, 0, -2$. In the three panels we fix $E_B \simeq 0.5 E_F$ and unbalanced polarization $N_\uparrow = 4, N_\downarrow = 3$ on a $4 \times 4$ square lattice. The lines dubbed $AP_\uparrow$ visible in the deep BEC regime of panels (a-c) come from the binding of the impurity with the extra unpaired spin up fermion, and are highlighted by the cyan dotted boxes.

to either a pair or the unpaired spin up fermion, so that the attractive lines labeled $AP_\uparrow$ are visible in panels (a-c), where we have also added the cyan dotted boxes as a guide for the eye. For $U_\uparrow = 0$, instead, the impurity can bind only to a pair, giving rise to only one attractive line, see panel (d).

## 6 Conclusion

To summarize, we have adapted the exact diagonalization method to the computation of polaron spectra on top of different many-body backgrounds, described by extended Fermi-Hubbard models. Varying the range and the sign of the interactions between the fermions and the polarization of the system, we have predicted that charge density waves, multiple Fermi seas and Fermi superfluids display polaron spectra with distinctive features. This supports polaron spectroscopy as an effective way of probing many-body systems.

As already mentioned, cold atoms offer a promising platform for the exploration of the results presented in this work. Mobile impurities can be injected in atomic quantum gases, by exploiting controlled transfer between different atomic internal states [4]. In this context, both inter-species and intra-species interactions can be tuned by means of Feshbach resonances [43, 74–76]; atomic mixtures with controllable imbalance can be prepared [77]; lattices of various geometries can be designed [78]; and polaron spectra can be obtained through RF

spectroscopy [4,79] and (momentum-resolved) Raman spectroscopy [80]. By manipulating small ensembles of atoms [41,42], these systems could explore the extrapolation between the ED results presented in this work and their thermodynamic limit. Optical lattices can also be engineered, even though, as we have shown in Appendix C, the presence of a lattice is not an essential physical ingredient to obtain our results (with the exception of Section 3, the lattice just makes it technically possible to perform ED). Possible limitations of these systems concern the contact nature of the inter-particle interactions and the existence of three-body recombination, which limits the measurement time and can be kept into account only phenomenologically in our theoretical calculations, via the width of the lines. An interesting perspective concerns polarons in dipolar gases exhibiting finite-range (dipole-dipole) interactions [39,81], as well as long-range interactions mediated by an additional Bose gas, which was recently proposed as a strategy to realize crystalline states [82].

Regarding solid-state realizations, intriguing quantum many-body phases, including the ones treated in this paper, have already been realized in moiré TMD heterostructures [15–26], and optical spectroscopy is a rather straightforward and well-established technique in this context [14]. The main limitation of polaron spectroscopy in current platforms concerns the fact that the binding energy of the trion is not easily tunable, but is generally of the order of 20-30 meV, hence larger than the typical electronic many-body gaps of few meV's. In particular, this explains why the avoided crossing in the repulsive branch predicted in Section 3 was not reported in the Wigner crystal experiments of Refs. [17,19,59]. However, we expect that trions with reduced binding energy may become accessible in the coming years, e.g. by a careful engineering of screening or building on the bilayer Feschbach resonance [83].

On the technical side, ED is limited to very small system sizes. Despite this limitation, comparisons with other approximate methods or experiments (whenever possible) suggest that our ED approach yields qualitatively solid results. An important theoretical puzzle, which is worth investigating in the future with complementary methods, concerns the description of the attractive polaron branch displayed in Fig. 5, in the case of the fermionic superfluid. Moreover, the ED results shown in Figs. 2 and 3 have pushed us towards developing a theory of polarons in charge density waves, an investigation which will be published soon. Other future efforts will be directed to computing polaron spectra in unconventional settings using the ED method, including topological baths and Rabi driven impurities [84–86].

# Acknowledgments

We would like to thank Atac Imamoglu, Jacques Tempere, Giacomo Mazza and Haydn Adlong for stimulating discussions. All numerical calculations were performed using the Julia Programming Language [87].

**Funding information**   This research was financially supported by the ERC grant LATIS, the EOS project CHEQS and the FRS-FNRS (Belgium). Computational resources have been provided by the Consortium des Équipements de Calcul Intensif (CÉCI), funded by the Fonds de la Recherche Scientifique de Belgique (F.R.S.-FNRS) under Grant No. 2.5020.11 and by the Walloon Region.

# A   Convergence analysis

We show in this appendix that it is possible, to a certain extent, to obtain consistent information about polaron spectra through different number of particles, system sizes and lattice shapes.

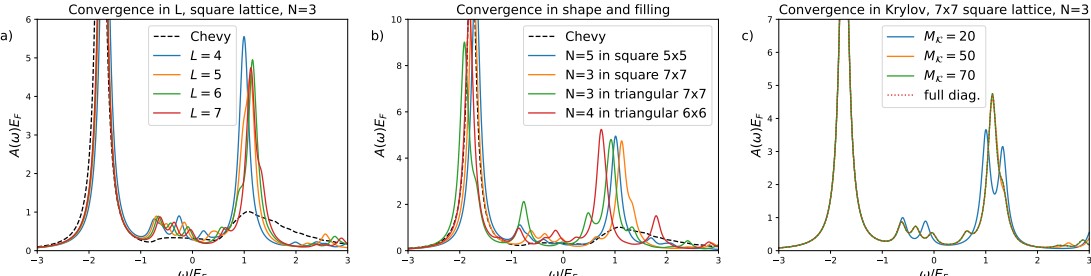

Figure 7: (a,b) Cut of Fermi polaron spectra at $E_B = E_F$, for different number of particles, system sizes and shapes. The black dashes correspond to a continuum Chevy calculation. Panel (c) shows convergence in the Krylov space dimension.

In particular, the attractive polaron looks like a very stable feature, while the particle-hole continuum is strongly dependent on the details of the simulation. The repulsive polaron is in between, in the sense that it is roughly the same through different numerical setups, but with some variability. As stated in the main text, our philosophy is that this moderate variability can actually be useful: if some feature (like the avoided crossing in the RP branch showed in Fig. 2 or the broadening in the AP of Fig. 5) is shared by different lattice shapes and sizes, it is most likely a physical result, because artifacts of lattice discretization and finite-size effects depend strongly on the numerical details.

In Fig. 7 we compare the spectrum of the polarized Fermi polaron at $E_B = E_F$ for different numerical setups. We also show convergence with the dimension $M_\mathcal{K}$ of the Krylov space by comparing with the spectral function obtained from the full diagonalization of the Hamiltonian. (In this particular case the Hilbert space has dimension $N_\mathcal{H} = 18424$ and so it is feasible, in the other scenarios investigated in the paper a full diagonalization is usually impossible. The largest Hilbert space considered in this work is for Fig. 5 and has dimension $N_\mathcal{H} = 5290000$.)

In black dashes we also compare with a Chevy computation for a continuum, fully polarized non-interacting Fermi sea $|FS\rangle$. More specifically, the Chevy Ansatz

$$|\Psi\rangle = \left( \psi_0 a_0^\dagger + \sum_{kp} \psi_{kp} a_{-k-p}^\dagger c_{k\uparrow}^\dagger c_{p\uparrow} \right) |FS\rangle \,, \tag{A.1}$$

leads to the equations of motion

$$i\partial_t \psi_0 = W(0)\langle N_\uparrow \rangle \psi_0 + \sum_{kp} W(k-p)\psi_{kp} \,, \tag{A.2}$$

$$i\partial_t \psi_{kp} = \left( \epsilon_k^f - \epsilon_p^f + \epsilon_{k-p}^I + W(0)\langle N_\uparrow \rangle \right) \psi_{kp} + W(k-p)\psi_0 + \sum_q W(q)(\psi_{k+q,p} - \psi_{k,p+q}), \tag{A.3}$$

where $W(q) = \frac{W_0}{A} e^{-\frac{\ell_W^2 q^2}{2}}$ is a very short-range fermion-impurity interaction with a Gaussian shape of size $\ell_W = 0.2 k_F^{-1}$ and depth $W_0$ chosen to yield $E_B = E_F$. The equations of motion are diagonalized on a polar grid with a momentum cutoff of $k_{\text{cut}} = 15 k_F$ (see Supplementary material of [31]). While all the spectral features have a very similar location in energy, it turns out that in ED the RP and molecule-hole continuum have a much large spectral weight. It is not clear whether this discrepancy is an ED finite-size artifact, an effect due to the microscopic lattice (so that in principle the two models are different), or, more intriguingly, a shortcoming of the Chevy approximation. For the RP branch in particular, multiple particle-hole scattering channels may be relevant.

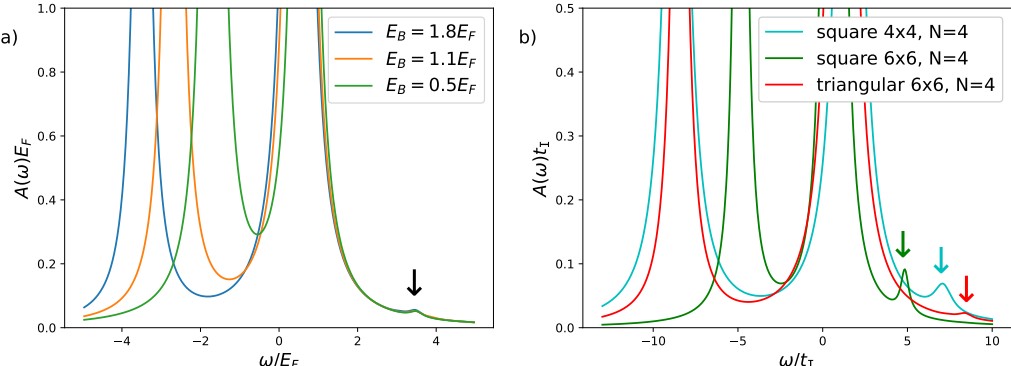

Figure 8: Umklapp peak phenomenology. (a) Spectral function for $N_\uparrow = 4$ in a $6 \times 6$ triangular lattice and three different fermion-impurity interaction constants. (b) Same for different microscopic lattices, with $E_B \simeq E_F$. In all cases, the strongly repulsive regime $V_0/t_I \sim 100$ is considered, for which one has a very rigid CDW.

## B  Zooming in on the umklapp peak

In this paragraph we report numerical evidence that the spectral feature labeled as umklapp peak in Section 3 behaves as expected. In particular, the umklapp peak energy is uniquely determined by the dispersion of the impurity and the lattice constant of the CDW, but will be (at first order) independent of the fermion-impurity interaction.

In Fig. 8(a) we display cuts of the spectral function for $N_\uparrow = 4$ strongly repulsive fermions in a $6 \times 6$ triangular lattice and three different $E_B$'s. While the attractive polaron branch is strongly affected, the location of the umklapp peak is mostly unchanged. In Fig. 8(b) we instead compare three different lattices at the same $E_B \simeq 1.8E_F$. Respectively, the single particle Bloch bands in an infinitesimal potential with the periodicity of the CDW suggests a RP-umklapp splitting of $6t_I$ for the $6 \times 6$ triangular lattice, $4t_I$ for the $4 \times 4$ square, and $3t_I$ for the $6 \times 6$ square, compatible with the spectra. Due to these large splittings (arising from the small CDW lattice constant), the oscillator strength of the umklapp peak is quite small, e.g. its height is 5% for the $4 \times 4$ square case.

We have also verified that the umklapp peaks do not shift with $t_f$ or $V_0$ (not shown).

## C  Results for square lattice

As already discussed in Sec. 2 and in Appendix A the polaron spectra on top of a non-interacting Fermi sea are, to a very good approximation, independent of the microscopic lattice. In this Appendix, we show that this is also the case for the repulsive spinful Fermi fluid described in Sec. 4 and the superfluid of Sec. 5. Since in the main text we reported results on the triangular lattice, here in Figs. 9, 10 and 11 we show the analogous computations for the square lattice. As the reader can see, the agreement is excellent, the only qualitative discrepancy being for the superfluid, in the behavior of the AP line at large $E_{\text{pair}}/E_F$, as already mentioned in the main text Sec. 5, where in Fig. 10 there is a secondary peak highlighted by the pink arrow, at lower energy than the main AP peak. This is also the case for the triangular $5 \times 5$ lattice with $N_\uparrow = N_\downarrow = 3$ calculation reported in Fig. 12. While this finite-size effect prevents us from making final statements on the precise structure of the AP peak, we can reasonably expect that the presence of some broadening in the AP branch or of a close-by secondary peak is a physical result; this is a novel feature which deserves further investigation

and the development of beyond Chevy theoretical tools.

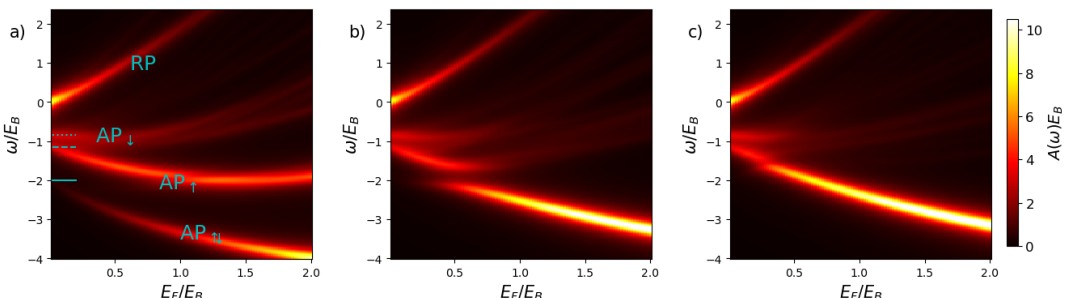

Figure 9: Same as Fig. 4, but for a $4 \times 4$ square lattice with $N_\uparrow = N_\downarrow = 3$ fermions, and $V_{\uparrow\downarrow} = 0, 2.4E_F, 4.8E_F$.

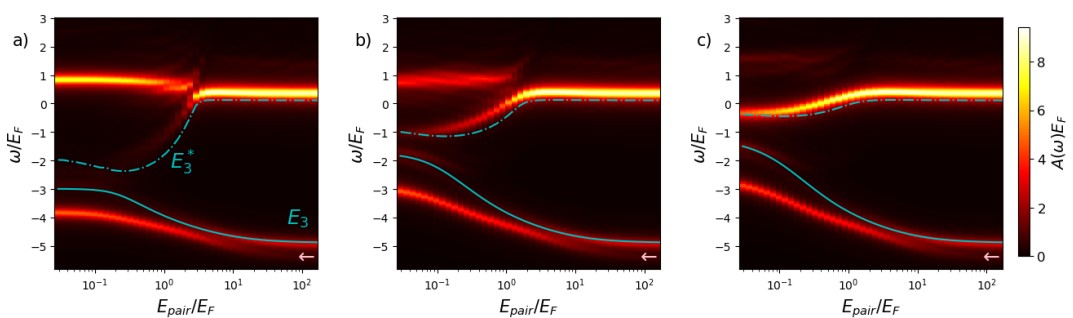

Figure 10: Same as Fig. 5, but for a $5 \times 5$ square lattice with $N_\uparrow = N_\downarrow = 3$ fermions, and $E_B = 0.5E_F$.

# D   Chevy Ansatz on top of BCS mean-field state

For comparison with the Fermi superfluid results in Sec. 5 of the main text, in this Appendix we compute spectra using the generalized Chevy Ansatz on top of the BCS mean-field state [29–31]. For notational simplicity, in the following we introduce the rescaled couplings $g = V_{\uparrow\downarrow}/L_x L_y$ and $g_\sigma = U_\sigma/L_x L_y$.

The BCS state $|BCS\rangle$ is defined as the vacuum (i.e. $\gamma_\sigma |BCS\rangle = 0$) of the quasi-particles

$$\gamma_{k\uparrow} = u_k c_{k\uparrow} + v_k c_{-k\downarrow}^\dagger, \qquad \gamma_{k\downarrow} = u_k c_{k\downarrow} - v_k c_{-k\uparrow}^\dagger, \tag{D.1}$$

$$c_{k\uparrow} = u_k \gamma_{k\uparrow} - v_k \gamma_{-k\downarrow}^\dagger, \qquad c_{k\downarrow} = u_k \gamma_{k\downarrow} + v_k \gamma_{-k\uparrow}^\dagger, \tag{D.2}$$

where $u_k = \sqrt{\frac{1}{2} + \frac{\xi_k}{2E_k}}, v_k = \sqrt{\frac{1}{2} - \frac{\xi_k}{2E_k}}$ are the Bogoliubov coefficients, with $E_k = \sqrt{\Delta^2 + \xi_k^2}$ the quasi-particle energy, $\Delta = -g \sum_k u_k v_k$ the gap and $\xi_k = \epsilon_k^f - \mu$ the bare dispersion shifted by $\mu$. The chemical potential $\mu$ is tuned to impose a given average number of particles $\langle N_\uparrow \rangle = \langle N_\downarrow \rangle = \sum_k v_k^2$, in this case chosen to match the same density as the ED calculation.

The Chevy Ansatz is obtained by exciting a quasi-particle pair out of the mean-field state

$$|\Psi\rangle = \left( \psi_0 a_0^\dagger + \sum_{kp} \psi_{kp} a_{-k-p}^\dagger \gamma_{k\uparrow}^\dagger \gamma_{p\downarrow}^\dagger \right) |BCS\rangle. \tag{D.3}$$

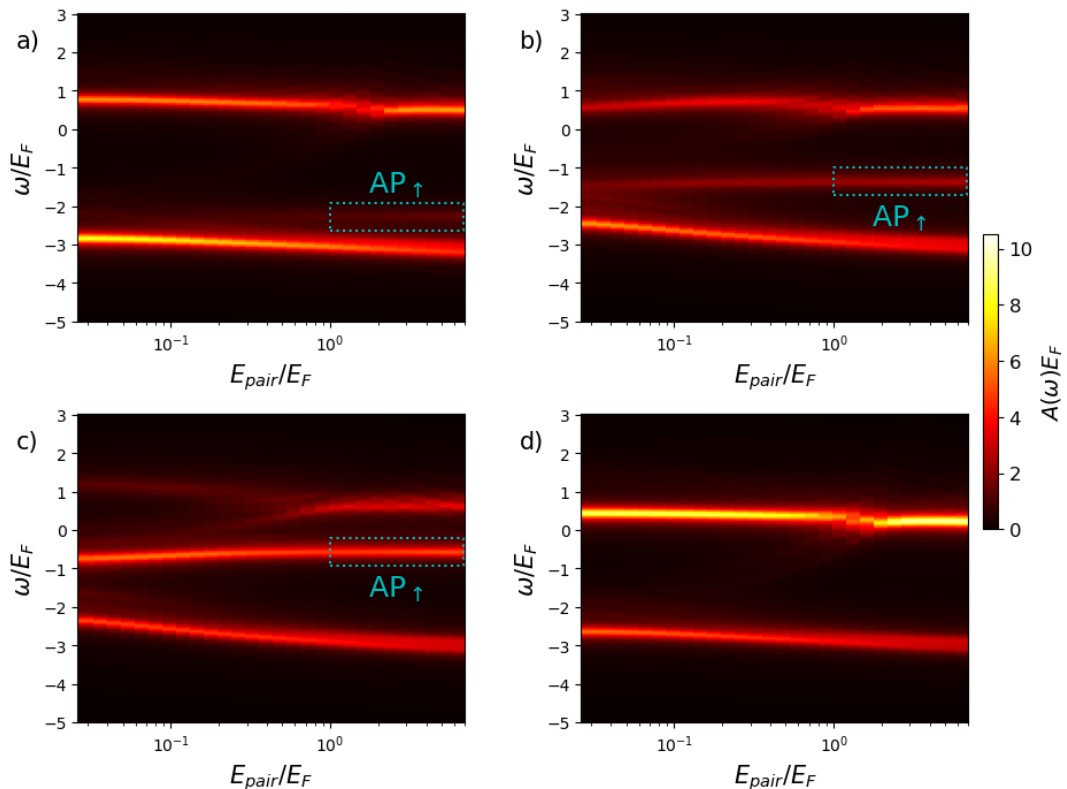

Figure 11: Same as Fig. 6, but for a $4 \times 4$ square lattice with $N_\uparrow = 5, N_\downarrow = 3$ fermions, and $E_B = 0.5E_F$.

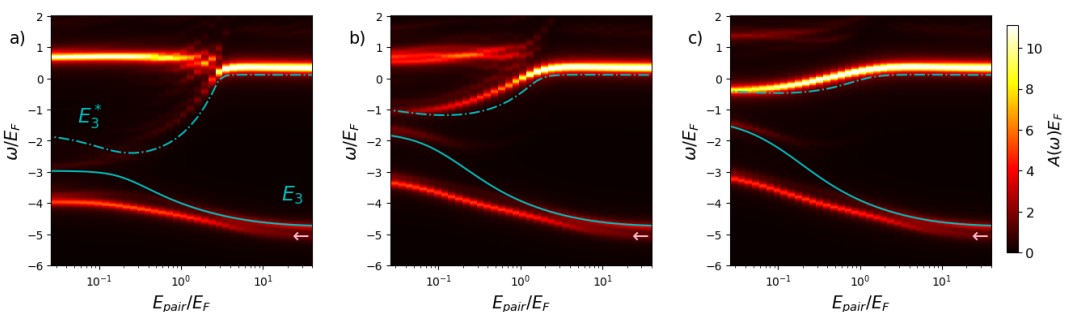

Figure 12: Same as Fig. 5, but for a $5 \times 5$ triangular lattice with $N_\uparrow = N_\downarrow = 3$ fermions, and $E_B = 0.5E_F$.

The Chevy-Schrödinger equations to be diagonalized numerically read

$$i\partial_t \psi_0 = (g_\uparrow \langle N_\uparrow \rangle + g_\downarrow \langle N_\uparrow \rangle) \psi_0 - \sum_{kp} (g_\uparrow u_k v_p + g_\downarrow v_k u_p) \psi_{kp}, \tag{D.4}$$

$$i\partial_t \psi_{kp} = \left( E_k + E_p + \epsilon^I_{k+p} + g_\uparrow \langle N_\uparrow \rangle + g_\downarrow \langle N_\downarrow \rangle \right) \psi_{kp} + (g_\uparrow u_k v_p + g_\downarrow v_k u_p) \psi_0$$
$$+ g \sum_q (u_k u_p u_{k+q} u_{p-k} + u_k v_p v_{k+q} u_{p-k} + v_k u_p u_{k+q} v_{p-k} + v_k v_p v_{k+q} v_{p-k}) \psi_{k+q,p-q}$$
$$+ \sum_{k'} (g_\uparrow u_k u_{k'} - g_\downarrow v_k v_{k'}) \psi_{k'p} + \sum_{p'} (g_\downarrow u_p u_{p'} - g_\uparrow v_p v_{p'}) \psi_{kp'}, \tag{D.5}$$

where the last three terms describe respectively up-down, impurity-up and impurity-down scattering processes. The spectral weight is given by the $|\psi_0|^2$ component of each eigenvector.

Differently from previous works, where (nonlinear) polar grids where used to achieve UV convergence, here we are directly interested in a lattice model. This automatically takes care of any UV issue, so that we can use Bravais grids for the momenta. As a consequence, the three interaction processes (up-down, impurity-up and impurity-down scattering) can be dealt with on an equal footing, and we can treat not only the $U_\uparrow = U_\downarrow$ [29] and $U_\downarrow = 0$ [30, 31] cases, but also all the intermediate ones.

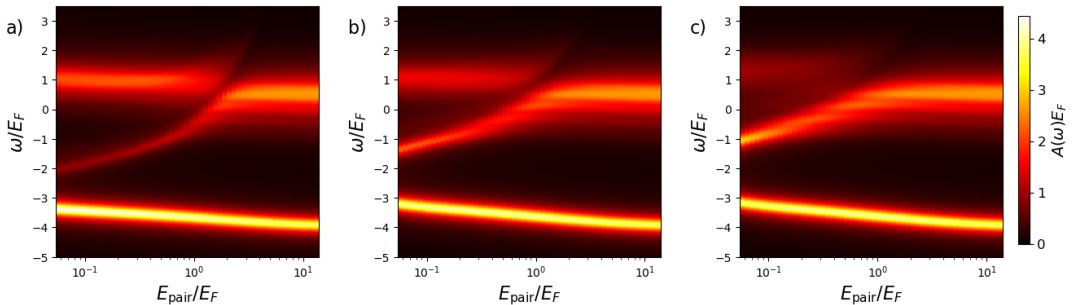

Figure 13: Polaron spectra for a Fermi superfluid from a Chevy calculation on top of a BCS ground state. The three panels correspond to $\frac{U_\uparrow - U_\downarrow}{U} = 2, 1, 0$, respectively. A triangular 11×11 lattice was used, with an average filling of one fourth, and impurity-fermion interactions of strength given by $E_B = 0.5E_F$. This plot shows remarkable analogies with its ED analogue Fig. 5.

The results are reported in Fig. 13. There is an excellent qualitative agreement in the repulsive branch phenomenology with the ED calculation. What is instead missing are the secondary contributions to the AP line at large $E_{pair}/E_F$, which may suggest an origin from Goldstone fluctuations, which are neglected in the Chevy subspace. A way of taking care of these processes in the polaron problem is an open theoretical challenge, and will be the subject of future research efforts.

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
