# Peer review of "Polaron spectroscopy of interacting Fermi systems: insights from exact diagonalization"

_SciPost Physics, doi:SciPost Phys. 16, 056 (2024)_

## Round 1 · Referee Report · Anonymous (Referee 1) · 2023-10-12

Strengths

1- A wide variety of polaron problems is studied, including many problems that have not been studied in detail yet using other methods. 2- These problems are studied in detail and compared with previously known results, in general finding good qualitative agreement.

Weaknesses

1- The authors discuss that finite size effects are strong, but no attempt is made to quantify them. For instance, is there any form of convergence of the polaron spectra for 3x3, 4x4, 5x5 to the continuum result? 2-Similarly, can some statement be made about how the discrete lines in Fig.2 converge to a molecule-hole continuum? 3- Also, how do the results depend on the filling of the lattice?

Report

The paper studies in detail how polaron spectra can be studied using exact diagonalisation, finding (to my surprise), that even small systems suffice to get realiable estimates for the qualitative features of polaron spectra. Due to the relevance of polaron spectra to a wide variety of fields in condensed matter and ultracold atomic physics, and the many directions that can be explored from here, I can recommend this paper for publication after the points in weaknesses are at least attempted to be answered in more detail.

Requested changes

1- Answer the questions in "weaknesses"

(optional): Polaron's have been recently also probed using Rabi oscillation spectroscopy, see for example https://journals.aps.org/prl/abstract/10.1103/PhysRevLett.125.133401 or https://arxiv.org/abs/2308.05746 or https://arxiv.org/abs/2308.06659. Rabi oscillation's are very challenging to calculate with field theory methods (due to conceptual difficulties), but should be easy to calculate in ED, so that could be an interesting application of this method.

---

## Round 1 · Referee Report · Anonymous (Referee 3) · 2023-10-17

Strengths

1) The authors present results for a bunch of different interacting systems. 2) Despite finite-size effects of exact diagonalization, the results show characteristic features of polaron physics. Thus the paper gives an important benchmark for further investigations.

Weaknesses

1) The method is limited to rather small system sizes. More "few-body physics" is examined than "many-body physics". 2) Not all the features in the polaron spectra are suffiently discussed and understood.

Report

The article “Polaron spectroscopy of interacting Fermi systems: insights from exact diagonalization” by I. Amelio and N. Goldman presents spectral functions of polarons immersed in differently interacting 2D Fermi-Hubbard lattices obtained from exact diagonalization.
By changing the parameters of the Fermi-Hubbard model, the authors treat four different scenarios. At a spin-polarized background, they first benchmark the method on a non-interacting Fermi gas and then add repulsive Coulomb interactions giving rise to the formation of charge density waves. Next, they use a background of spinful fermions, where they study the effect of local repulsive and attractive Hubbard interactions to simulate the physics of transition metal dichalcogenides (TMD) and a polaron immersed in a superfluid, respectively. Here, they allow for spin imbalance and spin-dependent coupling between the impurity and the Fermi gas.
Although the method is limited to rather small system sizes, i.e., 3-8 particles on square and triangular lattices of sizes 4x4 up to 6x6, their spectra show characteristic features of polaron physics, i.e., the attractive and repulsive polaron branches or the signatures of umklapp scattering in presence of Coulomb repulsion. This is a valuable insight for the polaron community. However, their spectra obtain other features like the double line of the attractive polaron branch in a superfluid, which are not adequately explained. It remains an open question whether these are finite-size effects or really give hints to new physics.
Considering this and the following comments and questions, I am not able to yet give a final recommendation for whether to accept this article. To this end, I would kindly ask the authors to address the comments and questions listed under "Requested changes".

Requested changes

1) Exact diagonalization typically comes with an extreme restriction in terms of the system size. In my opinion, in the abstract it should be made clearer that the considered system sizes are extremely small and the manuscript actually deals with few-body physics rather than many-body physics.
2) To get a better feeling of the numerical complexity, could the authors please specify how large is the considered Krylov space, i.e., what are typical values of M used in the computations?
3) The plots, especially those whose colors are scaled logarithmically, i.e., Fig. 1 and Fig. 2 (b), show a bunch of other lines beside the usual polaron branches. Have the authors investigated whether their number and shape is in relation to number of particles or lattice sites? Also, regarding Fig. 2 (b), how large is the strength of the peak related to umklapp scattering in comparison to the other peaks?
4) For the non-interacting gas and the charge density waves, the authors give results for both the square and the triangular lattice. However, the additional features in Fig. 3 (a) for a triangular lattice are not discussed extensively. Why is the repulsive polaron branch split in two at around $K = 10$ and why does the linearly rising second band have a bifurcation at around $K = 3$? TMD results, i.e., Fig. 4, are exclusively shown for triangular lattices whereas BCS results, i.e., Figs. 5 and 6, are only shown for square lattices. I do not follow the author’s strategy of when they show results for the square lattice and when for the triangular lattice. This is all the more confusing, as in typical TMD experiments the electrons predominantly explore the bands at the band minimum, i.e. for momenta where the dispersion relation is quadratic, and thus lattice effects (band warping etc.) does not matter. For a transparent and complete description, the authors should consider to present the other lattice’s results in an appendix or they just show results for the square lattice after showing once both in Fig. 1. Moreover, a discussion for why band effects should matter at all to simulate TMD physics of Fermi polarons should be included.
5) At the beginning of Sec. 4: I assume that K inside the quantum numbers (K,uparrow) etc. represents the momentum values k. I first confused it with the previously used interaction strength parameter $K = V_0/(a_\mathrm{CDW} E_F)$.
6) At the beginning of page 10, the authors remark that to identify the different lines for the attractive polarons in Fig. 4 they calculated the various binding energies. It would be more convincing if the authors show the binding energies in Fig. 4 as they already do that with cyan lines in Fig. 5.
7) In the Sec. 5 the authors compare their plots several times with those they got from a Chevy ansatz and mean-field calculations of polarons in a BCS superfluid. There, the spectra are given in terms of the density n instead of $E_F/E_B$. On Page 9, it is described that varying $U = (U_\uparrow + U_\downarrow) / 2$ corresponds to varying $E_F/E_B$ and thus is similar to varying the density. Maybe it is just me, but could you make this point clearer. What is the actual expression for the binding energies in terms of $U$? At the end of Sec. 2.1, the authors claim that it is “[their] philosophy th[e] method naturally needs to be complemented with other approaches, such as Chevy ansatz”, but for my taste there has not been made enough effort for a detailed comparison.
8) In the conclusion, it is mentioned that polaron spectroscopy is an effective way of probing many-body systems. Could the authors elaborate more on how the systems presented here can be realized in experiments? There is a great progress in simulating Hubbard models in optical lattices, also few-body physics is being examined in cold-atomic gases. Are there some specific challenges to simulate experimentally the scenarios used in this work?
9) There are two phrasings which are not really precise regarding physics:
a. Page 3: “non-interacting Fermi polaron” refers to polaron immersed in a non-interacting Fermi sea, however the impurity-fermionic interaction $U$ needs to be non-zero for a polaron to be formed.
b. Page 6: I stumbled over the phrase “the impurity binds to an object of effective mass $N_\uparrow m_f$, where $m^*_f = \hbar^2/(2t_f)$ is the bare effective mass of a fermion”. I guess you mean “$N_\uparrow m^*_f$” and “bare effective mass of a fermion” is an somewhat confusing expression for “the effective mass of a single fermion” opposed to the “bare mass of a fermion”.
10) The authors are usually very precise when it comes to units. The absorption spectra have a dimension of 1/energy. Are the plotted absorption spectra given in units of $1/E_F$?
11) Finally, there are some defects in orthography and presentation, which leaves the reader with the impression that not enough effort has been made in the final polishing of the draft (with decreasing relevance).
a. Fig. 4(c) is squeezed compared to their counterparts (a) and (b).
b. Figs. 5 and 6 do not provide a colorbar legend.
c. There is an inconsistent way of referring to figures: “panels (a-c)” vs. “panels a, b, c” vs. “panels (a) and (b)”
d. Obvious orthographic mistakes: “Feschbach”, “blushift”
e. Inconsistent use of hyphens: “Fermi-Hubbard” vs. “Fermi Hubbard”, “ground state” vs. “ground-state”, “finite-size effect” vs. “finite size effect”, “Chevy ansatz” vs. “Chevy-ansatz”
f. Specific orthography: “p.e.” is not an official abbreviation, please use “e.g.” instead, in American English “labelled” is written as “labeled”.
g. Usually if a section is not specified with a numbering the word “section” is written in lower case.

---

## Round 2 · Referee Report · Anonymous (Referee 1) · 2024-1-25

Report

My questions have been fully answered and I recommend publication as-is.

---

## Round 2 · Referee Report · Anonymous (Referee 3) · 2024-2-5

Report

I appreciate that the authors took the referees’ comments very seriously. With the careful and detailed modifications in the text and the additional appendices the results are presented in a much more transparent and convincing way.

The first appendix underlines the robustness of the attractive polaron branch with respect to system size. The second appendix nicely illustrates that the umklapp peak only depends on the lattice details, but not on the interaction. Also, I appreciate that the authors now show both results for the triangular and square lattice and give more details on the Chevy ansatz in a BCS superfluid. Generally, the arguments in the paper are much more solid and evident now.

The plots all include colorbars now and the units of the plotted quantities are always provided. I have finally understood the notation with the K-point in the Brillouin zone. Good that the authors now clarify that in the text!

I still have some minor orthographic comments: - The first sentence on page 14 is not correct: “the lattice just makes it technically possible to perform ED”. - Typos in App. D: “intrduce”, “quasi-particles” vs. “quasiparticles”. - Inconsistencies of references: “panels (a), (b), (c)” vs. “panels (a-c)”.

The presented analysis with exact diagonalization gives new insights into polaron physics. Above all, I find it rather striking that such small system sizes and the confinement to a lattice reveal typical many-body features. Hence, besides the mentioned trivialities, I recommend the article for publication.

---

## Round 2 · List of Changes

We thank the two Referees for their constructive remarks. Apart from a series of changes in the main text and related plots (that we discussed below), we have performed extensive additional studies, which are now presented in four new Appendices. We believe that this additional material, which nicely supports the main results of our work, constitute a sizable improvement of our manuscript.

Report 1

We thank the Referee for their appreciation of our manuscript and for their useful remarks. We have now added a new Appendix A, which addresses the questions of the Referee about convergence. The bottom line of this analysis is that by comparing different sizes, fillings and lattices it is indeed possible to distinguish the features that would also hold in a thermodynamically large system, from those that are artifacts of finite-size effects.

Regarding the study of Rabi oscillations in this framework: we agree that it may be a very interesting application of our method, and we comment on it in the outlines. However, since this would still require a (conceptually straightforward) rewriting of some parts of our code, and since this does not completely fit the main topic of “interacting fermions”, we have decided to leave it as a future direction.

Report 2

We thank the Referee for their reading of our manuscript and for their useful remarks. In the revised manuscript we adopt a more critical attitude towards the “double line of the attractive polaron branch in a superfluid”. Indeed, while we are inclined to believe that this feature could likely be a finite-size effect, it might still suggest a strong contribution of bosonic, Goldstone-like excitations at energies very close to the AP line (as motivated by the comparison with Appendix D).
We reply point by point to the their questions 1-11:

1) We now explicitly mention the finite-size effect issue: “While
our calculations are performed using an exact-diagonalization approach, hence limiting
our analysis to systems of few interacting Fermi particles, we identify robust spectral
features supported by theoretical results.”.

2) We now discuss this in Appendix A, Fig. 7.c.

3) We now address these issues in Appendix A (finite size corrections) and Appendix B (umklapp peak).

4) First of all, most of the results do not rely on whether the microscopic lattice is chosen to be a square or a triangular one: Indeed, as also pointed out by the Referee, only the band minimum matters.
Following their suggestion, we now mainly display triangular-lattice results in the main text, and moved the square-lattice results to the new Appendix C (showing excellent agreement).

However, we point out that there is a genuine difference between triangular and square lattices in the case of the charge density wave because they give rise to different charge order, with different symmetries. For this reason, we have decided to plot both results in the main text.

Concerning the extra feature in the triangular-lattice results shown in Fig. 3a: We are currently investigating the spectra obtained from a Hartree-Fock approach combined with the Chevy ansatz; while these results will be presented in a future publication, we hereby summarize our current findings: while this Hartree-Fock-Chevy approach can reproduce the main avoided crossing due to the increase of the quasi-particle gap, we find that it does not capture this extra feature. Since larger system sizes are prohibitive, we cannot confirm whether this is a ‘beyond HF+Chevy’ effect or a finite size effect. This limitation is commented in the text.

5) We now explicitly define the +/-K points and use \kappa for the interaction parameter.

6) To reassure the reader, we now added the binding energies in Fig. 4.a, which perfectly match the ED lines. However, as expected, this agreement is only found at small E_F/E_B, since there is a strong redshift due to the Fermi sea at large E_F (this is why we had decided not to plot the binding energy in vacuum in the original version). Notice that this is different from Fig. 5, since in that case it is E_pair that is varied.

7) We now improved the presentation and explicitly provide the relation between the two quantities, U and E_B. The binding energy of the impurity-spin up molecule is still given by Eq. (5), and analogously for the spin down one.
To strengthen and clarify the comparison with the Chevy results, we have now added a new Appendix D, devoted to the BCS+Chevy calculation with the exact same setting as in ED.

8) Now we briefly discuss the main advantages and drawbacks of the cold atom and TMD platforms. Following the Referee’s suggestion, we also point out that our approach is interesting in addressing the question of how the many-body limit is reached starting from few particles and increasing their number.

9) This has now been fixed.

10) We are now more careful when drawing the colorbars of the plots, with proper units 1/E_F.

11) We have fixed all these presentation issues.

We thank once again the Refeere for all their constructive and detailed comments.

---

## Editorial Decision

published